# The Genetic Basis of Obesity and Related Metabolic Diseases in Humans and Companion Animals

**DOI:** 10.3390/genes11111378

**Published:** 2020-11-20

**Authors:** Natalie Wallis, Eleanor Raffan

**Affiliations:** Anatomy Building, Department of Physiology, Development and Neuroscience, University of Cambridge, Downing Street, Cambridge CB2 3DY, UK; njw64@cam.ac.uk

**Keywords:** obesity, genetics, companion animals, metabolic disease, comparative genomics, dogs, cats, horses

## Abstract

Obesity is one of the most prevalent health conditions in humans and companion animals globally. It is associated with premature mortality, metabolic dysfunction, and multiple health conditions across species. Obesity is, therefore, of importance in the fields of medicine and veterinary medicine. The regulation of adiposity is a homeostatic process vulnerable to disruption by a multitude of genetic and environmental factors. It is well established that the heritability of obesity is high in humans and laboratory animals, with ample evidence that the same is true in companion animals. In this review, we provide an overview of how genes link to obesity in humans, drawing on a wealth of information from laboratory animal models, and summarise the mechanisms by which obesity causes related disease. Throughout, we focus on how large-scale human studies and niche investigations of rare mutations in severely affected patients have improved our understanding of obesity biology and can inform our ability to interpret results of animal studies. For dogs, cats, and horses, we compare the similarities in obesity pathophysiology to humans and review the genetic studies that have been previously reported in those species. Finally, we discuss how veterinary genetics may learn from humans about studying precise, nuanced phenotypes and implementing large-scale studies, but also how veterinary studies may be able to look past clinical findings to mechanistic ones and demonstrate translational benefits to human research.

## 1. Introduction

Obesity presents a major health problem in humans and companion animals alike. An estimated 39% of people were overweight or obese in 2016, a value nearly triple that recorded in 1975 and equating to over 2 billion adults [1]. Mirroring that trend are increases in pet obesity, with as many as 63% of cats [2] and 59% of dogs [3] reported to be overweight or obese. Obesity was declared an epidemic by the World Health Organisation (WHO) in 1997 [4] and similarly identified as a major threat to pet health by BSAVA and WSAVA [5].

Precise definitions of obesity are debated, but it is generally accepted that obesity is the unhealthy accumulation of body fat. What is not controversial is that obesity is a consequence of energy intake chronically exceeding energy output. Consequently, obesity has commonly been considered a ramification of poor self-control in people or of inept management by animal owners. However, considerable evidence now shows obesity is better regarded as a disease of disordered energy homeostasis in which a multitude of genetic and environmental factors can contribute to increasing body fat.

This review first examines the pathophysiology of obesity and the role of genetics in the disease, focussing on the wealth of evidence from human and rodent studies. We then review the genetics of obesity and related metabolic disease in companion animals. Finally, we consider opportunities for future research in companion animals that may improve understanding of both animal and human obesity.

## 2. Factors Contributing to Obesity

Recent, rapid increases in the prevalence of obesity have been caused by changes in activity and diet in the human population, and the same is likely true in animal populations [6,7]. However, it is clear that, although much of the human population is exposed to an “obesogenic environment” (with ready access to high calorie food and increasingly sedentary lifestyles), only a subset become overweight or obese. It is now well established that multiple factors including socioeconomic status, education level, and genetics are associated with whether a person is likely to develop obesity [8]. In companion animals, similarly diverse risk factors have been identified, with biological factors such as age, sex, and breed recognised alongside owner management factors in dog, cat, and horse obesity [9,10,11,12].

Acknowledgement of the multiple obesity risk factors is important, because it informs efforts to reduce obesity. Stigma against overweight people and parents of overweight children is well recognised [13,14], and the same is true for owners of overweight companion animals [15,16]. Such stigmas arise from the widespread view that weight gain is due to lax efforts to regulate food intake and exercise, either by a person, parent, or animal owner. It is important to acknowledge the risk factors beyond an individual’s control (such as genetics) in order to improve the effectiveness of obesity prevention and treatment programmes.

### Obesity Susceptibility Is Highly Heritable

Humans within the same environment, be that energy surplus or energy scarcity, display a highly heritable variance in body condition [17]. A wealth of data from twin and adoption studies, bolstered by later estimates of chip heritability in the era of high-density genotyping, supports that human obesity, indicated by body mass index (BMI), is a highly heritable trait. Heritability estimates range from 71–81% [17,18,19]. However, despite intensive efforts, the genes and mutations responsible for the majority of this heritability remains to be elucidated.

## 3. Studies of Monogenic Obesity Have Been Highly Informative

Obesity is usually a complex trait, with many genomic loci contributing incrementally to modulate an individual’s susceptibility. However, monogenic forms of obesity also exist with patients usually coming to attention, because they develop severe obesity early in life. Interrogating the genetics of these rare patients, in combination with research in rodent and cellular models, has been hugely informative in elucidating the molecular basis of the regulation of energy homeostasis and body weight. Early studies of patients with monogenic forms of obesity focussed on candidate genes chosen based on information from rodent models of obesity [6].

### 3.1. The Discovery of Leptin

In 1950, a rodent model demonstrating severe obesity was identified, and the gene responsible named the obesity (*ob*) gene [20]. A similar obesity phenotype due to a different gene was discovered in diabetic mice, named the diabetes (*db*) gene [21,22]. Subsequent parabiosis experiments in which mice of contrasting genotype were surgically joined to share a circulation led to the conclusion that *ob/ob* mice lacked a circulating factor that controls eating behaviour, whereas *db/db* mice possessed such a factor but were not able to respond to it [23,24]. In 1994, the circulating factor was pinpointed and its function delineated; a hormone called leptin, which is secreted from fat cells [25]. The *ob* and *db* genes were subsequently identified as the genes for leptin (*Lep*) and its receptor (*Lepr*), respectively.

Shortly afterwards, a frameshift mutation in the human leptin gene (*LEP*) [26] was identified in children with severe, early-onset obesity. The mutation caused congenital leptin deficiency that was successfully treated with recombinant leptin therapy (See Figure 1). Since this, many more severely obese individuals have been identified with mutations in *LEP* [27,28,29] and provided help with recombinant leptin. Such studies clearly justify the importance of genetic research and its translational significance for prevention and treatment of disease.

Patients with mutation in the human leptin receptor gene (*LEPR*) [30,31,32] who display extreme obesity have also been identified. However, recombinant leptin treatment in these patients is entirely ineffective, since they suffer from leptin resistance as opposed to leptin deficiency [33,34,35].

### 3.2. The Leptin–Melanocortin Pathway

We now understand that the primary effector mechanism for leptin’s action is in the central nervous system (CNS), where it activates the hypothalamic leptin–melanocortin pathway [37], a neuroendocrine signalling mechanism, which transmits a signal about the status of the body’s energy reserves to the brain and translates it into effector signals to promote optimal energy balance. In summary (See Figure 2), the hormone leptin is produced and released from adipose tissue. In the insulin-dependent fed state, insulin stimulates leptin release, and in greater amounts when energy reserves (in the form of fat stored in adipocytes) are larger [38,39]. In the brain, leptin acts on receptors in the arcuate nucleus (ARC) of the hypothalamus to activate proopiomelanocortin (POMC) neurons to produce the pre-pro-protein proopiomelanocortin (POMC). POMC undergoes proteolytic cleavage to produce a number of neuroactive peptides, the most important of which in regulating energy homeostasis are α- and β- melanocyte stimulating hormone (α-MSH, β-MSH). MSH peptides act on melanocortin receptor 4 (MC4R) expressed on second-order neurons of the paraventricular nucleus (PVN) of the hypothalamus, resulting in reduction in food intake and modified energy metabolism by interaction with multiple other pathways [17,40].

### 3.3. Disruption of Leptin–Melanocortin Signalling Leads to Obesity

Mutations that disrupt the leptin–melanocortin pathway are associated with severe obesity in both rodents and humans [41,42]. Dominant mutations in the proopiomelanocortin gene (*POMC*) are reported to cause severe, early-onset obesity in affected patients, often with other neuroendocrine features consistent with the diverse physiological roles of the multiple POMC-derived peptides [43,44,45,46]. Similarly, obesity has been reported due to mutations affecting prohormone convertase 1 gene (*PCSK1*), one of the enzymes responsible for proteolytic cleavage of POMC to its neuroactive derivatives, either singly or as part the more complex genetic condition Prader–Willi Syndrome [47].

Variants residing in the gene *SIM1* have also been associated with severe obesity and Prader–Willi-like syndromes [48,49], an effect attributed to this transcription factor’s integral role in development of the PVN, a hypothalamic nucleus, which is most notable as the major site of MC4R expression [48].

More common are mutations in the *MC4R* gene, which have been shown to cause both dominant and recessive forms of monogenic obesity [50,51,52,53] and are responsible for up to 6% of severe, early-onset obesity cases [46,54,55,56]. More recently, *MC4R* variants with less severe effects on receptor function have been shown to be major modifiers of obesity risk in the wider human population [57,58,59]. Notably, those data show that the genetic background against which the *MC4R* mutation occurs has a large influence on the penetrance of the obesity phenotype.

### 3.4. Other Causes of Monogenic Obesity

Variants affecting other biological pathways have also been identified as monogenic causes of obesity, with the majority related to CNS regulation of energy metabolism. Semaphorin 3 gene (*SEMA3*) variants are rare causes of severe early-onset obesity and affect energy balance through their role in melanocortin neuron development [60]. Brain-derived neurotrophic factor (BDNF) acts on its receptor tropomyosin receptor kinase B (TRKB), and there is increasing evidence that this signalling plays a significant role in sustaining equilibrium of energy balance in the brain [61]. Mutations affecting the concordant protein-coding genes (*BDNF/TRKB*) have been reported as causes of monogenic human obesity.

It is noteworthy that all the aforementioned variants primarily modify eating behaviour. The exception to this came with the discovery that rare variants in the Kinase suppressor of Ras 2 gene (*KSR2*) appear to cause obesity by affecting both energy intake and energy expenditure (although primarily via affecting the central control of energy metabolism) [62]. Reminiscent of this, a *CREBRF* variant that is common in Samoans but very rare in other populations, appears to be a major modifier of obesity risk by altering energy use in the body [63].

## 4. Common Human Obesity

Although cases of monogenic obesity exist, they account for only 5–7% of severe obesity cases [17,54,64] and much less of the obesity that develops later in life. Common obesity is a complex trait, caused by the additive effect of hundreds, possibly even thousands, of common genetic variants [65]. Genome-wide association studies (GWAS) have provided valuable insight into which loci, genes, and variants are responsible. Large, consortium-based studies involving hundreds of thousands of human subjects have been performed on quantitative traits such as BMI (the best available indicator of body fat percentage for large scale studies) and obesity-related traits such as waist-to-hip ratio (WHR, an indicator of where an individual’s body fat is stored) and measures of insulin resistance (and related metabolites), leading to identification of hundreds of quantitative trait loci [66] for obesity and related traits across the genome [66,67,68,69,70,71].

Despite those successes, just 3–5% of obesity heritability is explained by existing GWAS data [72]. The “missing heritability” of obesity [72,73] is hypothesised to be due to large effect rare variants yet to be identified: many loci of small/moderate effect too common to find with GWAS; non-additive genetic effects; and copy number variants (CNVs); among others [73,74,75,76]. Of those, the large-effect rare variant is particularly relevant to those studying veterinary species—selective breeding may lead to enrichment of variants of large effect within a breed, which are otherwise rare across the species overall [77]. Notably, an expanding body of evidence implicates the microbiome in obesity pathophysiology, and obesity-associated gut microbiota populations have been shown to be heritable [78]. Thus, an individual’s microbiome may also account for a fraction of the heritable component.

Nonetheless, identified obesity-associated loci can be used to generate estimates of individuals’ risk of developing obesity, known as polygenic risk scores (PRS), whereby a weighted effect score is generated as the product of allele count and effect score for risk variants. PRS have been proposed as clinical tools although their application is currently limited [79,80], in part because there is evidence that their validity may be limited in ethnic groups other than that in which they were originally derived. Nonetheless, PRS have been shown to be effective measures with which to stratify genetic obesity risk across a population, as in Figure 3. They have also improved our understanding of how background polygenic risk can alter trait penetrance in the presence of mutations with moderate–large effect on the trait, such as in *MC4R* [57], mentioned above.

### From GWAS to Function

Notably, a large proportion of genes implicated by human obesity GWAS are preferentially expressed in the CNS [67,81], providing further evidence for the critical role of the brain in controlling energy homeostasis. Many genes implicated in monogenic obesity have also been flagged in these large-scale obesity GWAS, including *LEPR*, *MC4R*, and *POMC*, although they are usually not the most prominent findings [67,82].

Moving from GWAS-associated locus to identify causative variants and their mechanisms of action is difficult. Some of the most common obesity-associated loci identified lack clear mechanisms by which variants exert their effects. A notorious example of this is the repeated and robust association of BMI with genetic variants lying within an intron of FTO α-ketoglutarate-dependent dioxygenase gene (*FTO*) also known as fat-mass- and obesity-associated gene (in mice). Those findings led to extensive efforts to characterise the role of this gene and provide evidence that *FTO* plays a part in regulating food intake. However, it later transpired that the focus on the closest gene was inappropriate and that obesity association for this locus is, at least in large part, due to altered regulation of a neighbouring gene, iroquois homeobox 3 (*IRX3*), which has an impact on peripheral adipocyte metabolism [83]. Although those findings have been challenged [84,85], it remains an example of how complex it has been to move from BMI locus to novel obesity biology.

## 5. Genetic Insight into Obesity Comorbidities and Metabolic Syndrome

Obesity exerts a huge toll on human health due to its attendant comorbidities including type 2 diabetes mellitus (T2D), cardiovascular disease (CVD), stroke, and non-alcoholic fatty liver disease (NAFLD), among others [86,87,88]. Development of these comorbidities is largely mediated by a constellation of clinical features commonly referred to as “metabolic syndrome” that include raised blood pressure, dyslipidaemia, increased triglycerides, and insulin resistance [89,90].

However, it is recognised that the severity of obesity does not necessarily correlate with the onset or severity of metabolic syndrome, and that some people remain metabolically healthy despite severe obesity [91,92,93]. Those differences occur not just between individuals but also between ethnic groups, leading to different recommended cut-offs for what is regarded as a “healthy” BMI in different populations. For example, a WHO expert consultation considered alternative BMI “cut-offs” in Asian populations given their high risk of T2D and CVD at BMIs considerably lower than previously recommended [94].

How obesity leads to its comorbidities has been intensively studied in humans and model organisms, informed by genetics. For instance, the relationship between obesity and T2D is reinforced by the presence of vertical pleiotropy with shared loci implicated in GWAS for both T2D and BMI. The causal relationship between obesity and its complications has been confirmed by Mendelian randomisation studies [95]. Similar suggestions and confirmation of cause and effect come from GWAS and Mendelian randomisation studies of BMI/insulin resistance and NAFLD [96].

More nuanced mechanistic insight has come from the study of rare patients with severe, early-onset, monogenic forms of insulin resistance. Studies of rare patients with mutations affecting the insulin receptor (INSR) or which interrupt signalling downstream of it have provided insight into the pathogenesis of insulin resistance and consequent dyslipidaemia or hepatic lipid accumulation in common obesity [93]. For instance, common variants in the gene encoding insulin receptor substrate 1 (*IRS1*), an important component of the intracellular signalling pathway, are associated with T2D, most likely by making affected individuals more susceptible to developing obesity-associated insulin resistance [97].

Similar value came from the study of patients with lipodystrophy, a rare condition in which there is complete or partial absence of adipose tissue. Recognition that the absence of fat was associated with early and severe insulin resistance and features of metabolic syndrome was central to establishing that the ability to store fat is critical to maintenance of normal metabolic function [98]. Those conclusions were later supported by analysis of genetic determinants of insulin resistance in the wider population, which provided evidence that variation in susceptibility to obesity co-morbidities are in part due to variation in individuals’ ability to develop and maintain healthy fat storage depots [70].

### Overview—Molecular Mechanisms Underlying Obesity Co-Morbidities

Together, human genetic studies and those in cell and animal models mean we now understand much more about how obesity causes disease. Much of those data support the theory of “limited adipose expandability”. In brief summary, it appears that the body can accumulate fat in a healthy manner only until it reaches the limit of adipose tissue’s ability to expand. Subsequently, adipose tissue dysfunction leads to inflammation and increased lipid flux, which in turn leads to ectopic lipid accumulation in non-adipose organs such as the liver and muscles. Combined with local and systemic inflammation, this leads to widespread insulin resistance, which in turn, results in increased insulin demand as the body attempts to maintain glucose homeostasis. Initially, insulin-producing β cells in the pancreas meet this demand but ongoing insulin resistance and ectopic lipid deposition in the pancreas itself can ultimately lead to β cell failure, hyperglycaemia, and T2D. Simultaneously, the altered lipid flux leads to dyslipidaemia and multiple mechanisms converge to cause hypertension [99,100].

Whether an individual develops obesity-associated co-morbidities may therefore be influenced by genes at many levels: a genetic predisposition to weight gain increases the chance of obesity developing; the ability to develop and maintain healthy fat stores appears to differ between individuals and across ethnic groups; if variants that perturb insulin signalling are present, they may promote the development of insulin resistance; and genes may also influence the ability of β cells to respond with adequate insulin where resistance develops. Consequently, in common disease, it is the net effect of many variants affecting such systems, combined with environmental influences, which determines whether obesity-related comorbidities develop.

## 6. Applying Current Knowledge to Study Companion Animal Disease

It is well established that obesity and its sequelae are heritable traits in humans and laboratory animals. The majority of obesity genes affect central regulation of food intake but notable examples whereby genes affect energy expenditure do exist. Genetic variance in susceptibility to obesity co-morbidities is well established and can be due to effects on a variety of body systems and processes.

Importantly, this wealth of information is available to inform investigations into companion animal disease. However, there is much yet to learn, and it is equally true that study of spontaneously occurring, companion animal models of obesity is a potentially valuable way to learn new biology relevant to all species. To date, studies of the molecular basis of obesity and associated disease remain limited in companion animals, with the predominance of small-scale and candidate genes studies somewhat limiting their value.

Available evidence suggests much commonality in the pathophysiology of obesity between humans, dogs, cats, and horses, but there are important differences in the incidence and severity of obesity-related co-morbidities between companion animal species, which likely reflect fundamental differences in physiology. Below, we consider in sequence the studies performed in dogs, cats, and horses to investigate the genetic basis of weight gain and, where relevant, obesity-related disease.

## 7. Canine Obesity Genetics

An estimated 34–59% of pet dogs in developed countries are overweight or obese [3,9,101,102,103,104]. Multiple comorbidities are associated with canine obesity, most notably: orthopaedic disease, exacerbation of breathing difficulties, and urinary incontinence, along with significantly decreased life expectancy [105,106,107,108]. Canine obesity is associated with increased blood pressure, dyslipidaemia, and insulin resistance, and this has been characterised as “obesity-related metabolic dysfunction” [109,110]. There is also some epidemiological evidence that obesity is a risk factor for developing canine, but this is not a recognised problem in clinical practice diabetes [111]. Other human obesity-related complications such as CVD and NAFLD do not have equivalent recognised presentations in canine practice. As with humans, studies suggest that weight loss leads to normalisation of many of the associated metabolic perturbations in dogs [109].

Evidence for the role of genetics in governing canine obesity susceptibility comes from the fact that obesity prevalence differs dependent on breed, with some breeds predisposed (e.g., Labrador retrievers, pugs, golden retrievers) and others resistant (e.g., greyhounds, whippets) [10,112,113]. Concordantly, differences in eating behaviour and food preference between breeds have been shown [114,115].

Although some of the variation in obesity susceptibility between breeds could be down to differences in “fashion” for each breed, the repeated finding of breed as a risk factor for obesity means genetic determinants are more likely the cause. Such variation in trait susceptibility between breeds is common in dogs due to the species’ unusual population architecture in which there is high diversity as a whole combined with high homogeneity within breeds, the product of population bottlenecks at breed formation and subsequent intensive selection for breed-specific traits [116,117,118]. Heritability of canine obesity is yet to be determined, but some have speculated that it will be reminiscent of high estimates in humans and other animal species [119,120,121,122].

The genetic basis of body mass has been investigated using GWAS in dogs, but those results cannot be regarded as indicative of obesity because of the wide variability in body morphology breeds. Instead, those study results likely better reflect the combination of height, length, musculature, and limb: trunk ratio [123].

To date, no GWAS of canine fat mass or related traits have been reported. In part, this may be related to the difficulty of accurately phenotyping obesity in dogs; the aforementioned wide variety of skeletal size and shape in dogs precludes using simple and commonly available measures such as weight to estimate fat mass. Quantification of adiposity can be done using sophisticated imaging or biochemical techniques, electrical impedance or ultrasound measurement of fat deposits, but those cannot practically be applied in large cohorts [124]. The clinically widely used and well-validated alternative is to assign dogs a “body condition score” (BCS). This should be performed by a veterinary professional who uses a variety of haptic and visual clues to assign dogs to one of a series of ordinal scores. The most widely used 9-point scale [125] has been thoroughly validated [124,126] such that each one point increase in BCS above “normal” equates to approximately 10% increase in fat mass. A 5-point scale is also available and has been reported in some of the studies below.

### 7.1. Genes Investigated in Canine Obesity

Several candidate gene studies have been performed in domestic dogs and closely related, farmed *Canidae* species. The merit of looking at genetic factors in other *Canidae* species is arguable, but they are reported here, since previous studies have found them relevant. The best characterised of the later mentioned studies are the ones that have considered genes implicated in the leptin–melanocortin signalling pathway. All gene variants discussed can be found in Appendix A.

#### 7.1.1. POMC

As discussed above, POMC has an integral role in leptin–melanocortin signalling, a neuroendocrine pathway highly conserved across species. Raffan et al. [7] identified various canine *POMC* variants including a 14 bp deletion at position 17:19431807-19431821, present in Labrador retrievers and flat-coated retrievers (FCRs) but absent from a wide variety of other breeds tested. The deletion was shown to be a major modifier of weight, adiposity (measured as BCS on a 9-point scale), and appetite (measured using a validated, owner-reported measure of eating behaviour [114]) in ≥210 Labrador retrievers, with findings for weight and appetite replicated in ≥196 FCRs. Appropriate confounders were accounted for by adjustment for age, sex, neuter status, and colour, thus association results can be confirmed with high confidence. Notably, the mutation was more common in Labrador retrievers working as assistance dogs compared to the pet population. For each additional allele carried by either breed, there was an approximately 0.5 BCS/2 kg increase in adiposity/body mass, and a similar incremental increase in food motivation.

The authors showed the *POMC* deletion results in altered production of β-MSH and β-endorphin, two neuroactive peptides derived from *POMC*, and confirmed that canine MC4R receptors have similar affinity for and response to α-MSH and β-MSH as the equivalent human pairings. Whilst the reduced action of β-MSH on MC4R in the leptin–melanocortin signalling pathway is the most likely mechanism resulting in this phenotype, β-endorphin may also be implicated.

Subsequent studies have confirmed the presence of this mutation [120,127,128]. In the British Labrador retriever population, Davison et al. [127] found there was no association for the deletion with occurrence of diabetes mellitus (DM) in the breed (a finding that is not surprising given that obesity-associated insulin resistance is not clinically recognised as predisposing dogs to diabetes, despite there being some epidemiological evidence that it may play a role population-wide).

The data had relevance to those studying human obesity biology, because canine *POMC* is highly homologous to human *POMC*. The same cannot be said for rodents that lack a proteolytic cleavage site in the pro-protein and so do not produce β-MSH. Since humans with mutations causing β-MSH absence are very rare [129,130], the canine model in this case provided important insight into the role of this particular peptide in energy homeostasis, as well as establishing a clear role for genetics in governing obesity susceptibility in dogs.

#### 7.1.2. MC4R

Canine *MC4R* gene polymorphisms have been identified in several studies, with some attempting to analyse their effects on obesity-related phenotypes. *MC4R* is a suitable candidate gene due to MC4R’s central position in the leptin–melanocortin pathway and given *MC4R* variants are among the most common cause of monogenic human obesity. Humans with *MC4R* mutations are also modestly taller than unaffected individuals of the same age and sex, or similarly obese unrelated subjects without *MC4R* mutations [53,131]. The majority of the identified variants lack functional data for characterisation of their consequences, but Yan et al. [132] reported that the missense variant c.637G>T (p.Val213Phe, rs852614811) had no significant effect on cAMP-mediated signalling downstream of the receptor.

Skorczyk et al. [133] conducted a study on canid *MC4R*, mapping and characterising it for the first time. The study cohort incorporated 31 dogs of 19 breeds and 35 farmed red foxes (*Vulpes vulpes*). They identified three polymorphic sites in the canine *MC4R* gene and four in the red fox. The three canine *MC4R* polymorphisms identified were a non-synonymous variant (c.637G>T, p.Val213Phe, rs852614811); a synonymous coding variant (c.777T>C, p.Ala259Ala, rs851987283); and a 3′UTR variant (c.*33C>G, rs851539399). No association testing was conducted.

Van den Berg et al. [134] studied *MC4R* in 32 golden retrievers. They identified the same polymorphic sites as Skorczyk et al. [133] (c.637G>T, p.Val213Phe, rs852614811; c.777T>C, p.Ala259Ala, rs851987283; c.*33C>G, rs851539399), plus an additional synonymous coding variant (c.868C>T, p.Leu290Leu, rs851062983). After linkage disequilibrium (LD) filtering, they performed an association study for the three remaining polymorphisms (c.637G>T, c.777T>C, c.868C>T) in a larger cohort of golden retrievers (*n* = 187). Several morphological measures were tested: weight, length, height, and body index score (BIS), but not BCS. For the association, they appropriately corrected for sex, age, and polygenic effects. However, they do not mention any account for neuter status (a well-recognised risk factor), and there was no measure of relatedness in the sample. No statistically significant association was found for any of the phenotypes. Whilst this may be genuine, the study was underpowered to find a small effect size, and these variants may warrant further investigation.

Zeng et al. [135] conducted an *MC4R* candidate gene study in beagles from a research colony. By sequencing two beagle dogs, they identified the previously reported synonymous coding mutation (c.777T>C, p.Ala259Ala, rs851987283) and a novel missense substitution (c.302C>A, p.Thr101Asn). (Note that the way this mutation is referred to in the original paper and a subsequent review article is somewhat confused. Zeng et al. [135] mis-name the mutations as C895T (for c.777T>C) and A420C (for c.302C>A). The missense mutation (c.302C>A) results in an amino acid substitution from threonine to asparagine (p.Thr101Asn = T101N), which can be clarified by figures within the paper. However, they misname this as N101T. Mankowska et al. [136], in a subsequent review, note the mistake but further misname it c.301A>C (p.Asn101Thr). The authors’ (unpublished) capillary sequencing data means we can confirm the correct annotation as c.302C>A, p.Asn101Thr. Variant nomenclature corrections can be found in Appendix A). In 120 beagles, c.302C>A was significantly associated with body weight (*p* < 0.05). In contrast, c.777T>C was only significantly associated with body weight in the heterozygous (but not homozygous) state and in bitches only. Whilst it is plausible at least one of the single nucleotide polymorphisms (SNP) are associated with adiposity, there are important limitations to the study. Specifically, weight does not equate to adiposity but could indicate changes in height, length, or muscle mass. There was no adjustment for confounding factors such as age, sex, or neuter status, and no assessment of whether the variants were in LD and therefore whether these associations constitute independent signals.

Raffan et al. [7] sequenced the *MC4R* region in 33 Labrador retrievers of which 15 were obese and 18 were lean. Two novel *MC4R* variants were identified (c.989G>T, p.Ser330Ile; c.*227C>T) plus three previously reported variants (c.637G>T, p.Val213Phe, rs852614811; c.*33C>G, rs851539399; c.777T>C, p.Ala259Ala, rs851987283). None were distributed differently between the small lean and obese groups, and the variants were not pursued further. This group also sequenced *AGRP* in this cohort, another valid candidate gene which codes for agouti-related protein—a neuropeptide known to modulate food intake in the ARC [137]. No *AGRP* variants were identified.

Mankowska et al. [136] investigated *MC4R* variants in 270 dogs of four breeds in which they identified six known polymorphic sites (c.637G>T, p.Val213Phe, rs852614811; c.777T>C, p.Ala259Ala, rs851987283; c.868C>T, p.Leu290Leu, rs851062983; c.*33C>G, rs851539399; c.*227C>T; and c.-435T>C, rs852471376). Of the 270 dogs, they had full phenotypic data for 164. They concluded that none of the identified variants displayed differential association with BCS (5-point scale) or body weight. The study population was dominated by Labrador retrievers (*n* = 187) and no potential confounding factors were accounted for.

#### 7.1.3. FTO

The top association signal on most human GWAS for BMI, *FTO*, is of debatable merit as a candidate gene for canine obesity, due to the controversy as to whether *FTO* itself or neighbouring genes are the true effector pathway causing the association signal (see above). Grzes et al. [138] investigated *FTO* SNPs in four *Canidae* species including the dog. In 39 dogs of 14 breeds, sequencing identified six polymorphic sites in *FTO*: one missense variant (c.23C>T, p.Thr8Met, rs852870212 described as “23 C/T, Thr1Met”), two intronic variants, and three 3′ flanking variants. Presence of the missense mutation differed by dog breed, but the small number of dogs genotyped meant association tests were (appropriately) not performed. However, of seven *FTO* polymorphisms in the red fox, one (a 5′ flanking region variant) was tested for association with body weight and pelt weight (which the authors report is affected by subcutaneous fat mass) in a larger cohort of 390 red foxes. No significant association was identified.

Grzemski et al. [139] used targeted next generation sequencing (NGS) in 32 Labrador retrievers in a region incorporating *FTO* and *IRX3*. Several polymorphisms were identified and tested for association with BCS (5-point scale) in a larger Labrador retriever cohort (*n* = 165), suitably adjusted for sex, age, and multiple testing. No association was found. The authors reported 56% nucleotide identity between human and canine *FTO* and 72% for *IRX3*. Additionally, they compared methylation status of CpG islands between lean and obese dogs in a smaller cohort (*n* = 28)—no differences were found.

#### 7.1.4. MC3R

Skorczyk et al. [140] conducted a candidate gene study on the gene encoding melanocortin receptor 3 (*MC3R)*. The *MC3R* gene is a reasonable candidate, although it is notable that rodent and human phenotypes associated with *MC3R* variants are more subtle than for the related *MC4R,* with *MC3R* most likely affecting the maintenance of body mass within the homeostatically controlled upper and lower limits [141,142,143,144].

The study cohort used by Skorczyk et al. [140] consists of four canid species: 31 dogs of 19 breeds, 35 red foxes, 30 arctic foxes (*Vulpes lagopus/Aloplex lagopus*), and 30 Chinese raccoon dogs (*Nyctereutes procyonoides procyonoides*). Multiple polymorphisms were found in *MC3R* in three of the species; no variants were identified in *MC3R* of the arctic fox. Two polymorphic sites were identified in the dog: short tandem repeat (STR) in 5’ flanking region (c.-90delT, rs853092001) and a synonymous substitution (c.142C>T, p.Leu48Leu, rs8916554). Association analysis for *MC3R* variants was then performed in cohort of 376 male red foxes. Two SNPs were in LD and both were associated with a small, statistically significant increase in body mass. There was limited information provided on potential environmental confounders and population stratification, with no apparent adjustment. It therefore cannot be ruled out that any associations observed may be as a result of these factors.

#### 7.1.5. INSIG2

Insulin-induced gene 2 protein (INSIG2) has a role in lipid metabolism and has been linked in human GWAS studies to various measures of circulating blood lipids. Its coding gene (*INSIG2*) has been implicated in human GWAS for BMI [145], but that association has not been reliably replicable [146], so its validity as a candidate obesity gene is somewhat limited. Grzes et al. [138] analysed polymorphic sites in *INSIG2* in four *Canidae* species including the dog. In 32 dogs of 14 breeds, seven polymorphic sites were identified: two 5′ UTR variants referred to as 5′ flanking region variants, (c.-90G>A (referred to as–91 G/A), rs852813691; c.-1C>T, rs852335828), one missense variant (c.40C>A, p.Arg14Ser, rs850773724), and four intronic variants. No association test was conducted in dogs, but one intronic SNP identified in the red fox was significantly associated with pelt weight in 390 red foxes. Skin weight may be influenced by subcutaneous fat mass, but this is not a valid measure of adiposity and such findings in the red fox may not be translatable to dogs.

#### 7.1.6. GPR120/FFAR4

G-protein coupled receptor 120 (GPR120), also known as free fatty acid receptor 4 (FFA4/FFAR4), functions as a receptor for unsaturated long-chain free fatty acids. The gene encoding this protein is known as *GPR120*/*FFA4/FFAR4* [147,148], and coding mutations in the gene have been associated with human obesity [149]. The following variants are described based on alignment to the canine *FFAR4* gene. Miyabe et al. [150] found nine *FFAR4* polymorphisms in a cohort of 141 dogs of 21 breeds: five synonymous substitutions (c.252C>G, p.Ala84Ala, rs852631320; c.282C>G, p.Pro94Pro (referred to as p.Asp94Asp), rs851850900; c.702A>G, p.Thr234Thr; c.726G>A, p.Thr242Thr; c.984T>C, p.Asn328Asn, rs852472019) and four non-synonymous substitutions (c.287T>G, p.Leu96Arg; c.307G>A, p.Ala103Thr; c.446G>C, p.Gly149Ala; c.595A>C, p.Thr199Pro, rs853030954 (referred to as c.595C>A, p.Thr199Pro)). SNPs were tested for association with BCS, the frequency of c.595A>C (referred to as c.595C>A) was significantly higher in dogs with a higher BCS. However, several confounding factors are unaccounted for and the study failed to correct for population stratification. Therefore, such identified associations may have been due to confounders or, given the unequally represented multi breed cohort, due to unaccounted-for population stratification.

#### 7.1.7. PPARs

Peroxisome proliferator-activated receptors (PPARs) are a group of nuclear receptor proteins that function as transcription factors and have roles in the regulation of cellular differentiation, development, and metabolism [151,152]. Nishii et al. [153] sequenced the genes encoding PPARβ and PPARγ (*PPARB* and *PPARG*) in two dogs and observed tissue-specific expression of various PPARs. They also investigated the presence of polymorphic sites in *PPARG* and identified a single polymorphism. No association test with phenotype was conducted. Since a relatively small canine cohort were genotyped, it is possible that other *PPARB* polymorphisms were missed.

#### 7.1.8. Adipokines

Adipose tissue communicates with the rest of the body in part by release of a range of molecular signals known as adipokines. Adipokines include leptin, pro-inflammatory molecules such as TNFα and IL6, and the much-debated resistin, the latter three of which were investigated in dogs by Mankowska et al. [154]. The choice of these genes is a little surprising in that, although there is evidence that each has a role in the development of insulin resistance secondary to obesity, there is little evidence that variants promoting inflammation are causal in obesity [155] and considerable evidence to the contrary (https://www.ebi.ac.uk/gwas/) [156]. Nevertheless, in 77 dogs of 17 breeds, Mankowska et al. identified multiple variants, including 13 in *TNF*, four in *IL6*, and eight in *RETN*. Three of these variants were missense substitutions: one in *TNF* (c.548A>T, p.Glu183Val) and two in *RETN* (c.19C>T, p.Leu7Phe, rs852470997; c.236C>G, p.Ser79Cys, rs851766760—referred to in the paper as a synonymous coding variant). The five most common variants (*TNF*: c.-40A>C, rs22216187; c.233+14G>A; *IL6*: c.309+215T>C; *RETN*: c.194-69T>A, rs853182485; c.75G>A, p.Glu25Glu, rs852185407) were genotyped in 260 dogs and tested for association with BCS, using breed-specific sub-groups to do the association analysis, including an “others” category for the poorly represented breeds. No association was found for the *IL6* and *RETN* variants with BCS in any breed group. The two *TNF* SNPs (c.-40A>C and c.233+14G>A) were significantly associated with body condition in Labrador retrievers but not in any other breed group. Whilst those associations may be meaningful, the failure to include recognised risk factors such as age, sex, and neuter status or to detect or correct for population stratification in the sample could have affected these results.

## 8. Feline Obesity and Associated Disease

Estimates suggest 12–63% of pet cats are overweight or obese [2,157,158,159,160,161,162]. Feline obesity is associated with multiple comorbidities, most notably DM and hepatic lipidosis [12,158,163,164,165,166,167]. This arguably makes human and feline obesity comorbidities closer compared to dogs [168,169]. However, human comorbidities such as hypertension and atherosclerosis are not commonly observed in obese cats [107,170,171].

Obesity’s relationship to DM in cats is well characterised with evidence suggesting a similar pathophysiology to human T2D [107,171,172,173] in which obesity leads to insulin resistance and ultimately β-cell dysfunction and diabetes [166,173,174,175,176,177]. Cats are therefore considered a suitable animal model for study of obesity associated T2D, with translational significance to humans. In contrast, whilst obesity is a well-recognised risk factor for feline hepatic lipidosis, the pathophysiological link between the two is less well characterised and subject to some debate [176,178,179].

### 8.1. Evidence for the Role of Genetics in Feline Obesity and Related Disease

Multiple studies suggest breed as a risk factor of feline obesity, with certain breeds displaying predisposition to becoming overweight/obese [12,180,181,182]. Although breed-specific obesity risk differs by study, domestic shorthaired cats (DSH) are consistently found to be at high risk, whilst longhaired breeds are generally at lower risk. Persian cats also have low obesity risk in most studies [12,181,182], but one study found them to be at high risk [180]. Together, these data suggest obesity may be at least in part a heritable trait in cats.

Notably, pet cats are made up of a majority of mixed breed cats (commonly referred to as DSH and domestic long-haired (DLH) breeds but are most commonly outbred and relatively genetically diverse) and a minority of pedigree cats, in which genetic architecture is reminiscent of dogs, with low within-breed diversity and evidence of population bottlenecks and genetic selection [183,184].

Until recently, genetic studies in cats have been stymied by the absence of a well-annotated, complete feline genome and lack of a commercial feline SNP genotyping array. Consequently, there are fewer genetic studies reported in cats overall, and none of the common forms of feline obesity. However, one group has reported candidate gene studies for feline DM [185], a related trait and one that is likely to be subject to similar genetic influences (including overlap with obesity predisposition) as described above for humans.

### 8.2. Familial Obesity in a Feline Colony

In a population of well-characterised, related research cats, a familial form of obesity has been reported [186]. Some cats displayed a clear predisposition to obesity, and segregation analysis suggested a single major gene was likely responsible. The report is more akin to human monogenic obesity or mutations of large effect against a variably obesogenic polygenic background.

A subsequent study found that the cats predisposed to being obese had a lower energy requirement and higher food intake than cats that did not tend to gain weight [187]. However, the energy requirement measurements were (for the obesity-prone group) performed not long after a period of restricted feeding and weight loss, interventions well recognised to cause reduced energy expenditure irrespective of baseline status [188]. In a subsequent generation of the same cohort, food intake and energy expenditure were investigated [189], and obesity-prone cats had higher food intake early in life but not lower energy expenditure.

In the same feline cohort, Keller et al. [190] investigated metabolic responses to different diets in cats predisposed to obesity vs. lean cats. No difference in metabolic response was found between the two groups. Additionally, in conference proceedings [191] the same group report attempts to map obesity genes were made and identified plausible candidate genes. Further comment is not possible given the scant information reported.

### 8.3. Genetics of Diabetes Mellitus in Pet Cats

Forcada et al. [185] investigated whether *MC4R* polymorphisms were associated with diabetes in DSH and Burmese cats. For each of a non-diabetic control group and a diabetic case group, there were 60 lean and 60 overweight cats, making 240 in total. The authors report that one *MC4R* polymorphism (c.92 C>T, p.Leu31Pro, rs783632116) was significantly more common in obese diabetic cats than obese non-diabetic cats, a finding not replicated in lean cats. Heterozygous and homozygous carriers in the non-diabetic subgroup were merged (assuming a dominant mode of inheritance). The authors do not report comparison of allele frequencies between lean and overweight cats (irrespective of DM status).

The authors speculate that this variant may act independently of an effect on body weight, but these reviewers suggest that to be a bold statement given the limitations of the data presented and the weight of evidence from other species about the role of MC4R in controlling food intake and obesity predisposition. For example, although *MC4R* is significantly associated with T2D in human GWAS, that association disappears when the analysis model corrects for BMI [192,193]. Consequently, it seems equally or more plausible that the association reported may in fact be a result of vertical pleiotropy rather than a direct effect.

The same group in conference abstracts report the results of a GWAS for feline DM in a cohort of 581 DSH cats [194,195], later adding Burmese cats [196]. However, the data are not reported in sufficient detail to reiterate here.

## 9. Obesity and Related Metabolic Disease in Horses

As in other companion animal species, equine obesity is common with an estimated 20–70% of horses overweight/obese [11,197,198,199,200]. This is a significant clinical problem, because obesity is a risk factor for the development of laminitis, a common, crippling disorder of the equine hoof. That association is thought to be mediated predominantly via a collection of risk factors known as equine metabolic syndrome (EMS). EMS was first described in 2002 [201], and the pathophysiological links between obesity and laminitis have been extensively studied since, although it is acknowledged that the current working understanding requires refinement [202].

EMS is defined by the presence of insulin dysregulation, characterised by clinical features including hyperinsulinaemia (either at baseline or in response to glucose challenge), hyperglycaemia, and/or evidence of peripheral insulin resistance [202]. Insulin resistance is commonly present in EMS, although there has been some debate as to whether that is always true and if alternative routes by which insulin dysregulation may develop exist [203]. Notably, although obesity and EMS are common, it is rare for horses and ponies to become diabetic [202].

Not all overweight equines develop EMS, nor does EMS always cause laminitis. Similarly, not all horses that have clinical features of EMS are overweight. Those paradoxes exist between individuals and across breeds, with some breeds apparently particularly prone to developing laminitis despite only moderate weight gain [198]. This is reminiscent of the situation described above in humans, whereby there is variability between individuals and ethnic groups concerning whether, and at what point, obesity-associated complications occur. It is clear, therefore, that a better understanding of equine obesity and its related conditions is required.

### 9.1. Genetics Influence Equine Obesity, EMS, and Laminitis

Evidence that genetics influence the development of equine obesity come from recognition that breed and “type” are clear risk factors with ponies at highest risk, followed by cob type breeds [11,204,205,206,207,208]. In Pura Raza Español horses, the heritability of BCS was found to be 14–24% [209]. Importantly, a within-breed study can only estimate the variance due to genetic variation within that (homogeneous) breed, so this heritability figure should not be generalised to the species as a whole. To date, there are no reports of GWAS or candidate gene studies in equine obesity.

The genetics of EMS have been more intensively interrogated. Breed is a well-recognised risk factor and breeds largely overlap with breeds at high obesity risk, unsurprising given the association between the conditions [202]. A similar variability between breeds has been demonstrated for insulin sensitivity and related biochemical parameters [210,211,212,213]. A recent study used genome-wide SNP data to estimate heritability of several traits known to be perturbed by EMS (glucose, insulin, measures of insulin sensitivity, and dyslipidaemia) and found they were moderately to highly heritable [214]. Again, such within-breed comparisons are valuable but are likely to underestimate the true heritability in the equine population.

Genetic studies of laminitis are worthy of report, given the common co-segregation of obesity, EMS, and laminitis. In an early study of crossbreed ponies, the authors concluded laminitis was a dominant trait in the pedigree with variable penetrance due to sex, age, and epigenetic-related variables [215]. Subsequent GWAS are mentioned below.

### 9.2. GWAS for EMS and Related Traits

Lewis et al. [216] performed a GWAS for laminitis and related traits in a population of 64 Arabian horses. A locus on chromosome 14 was associated with laminitis and insulin concentration. In a second cohort of 50 horses of the same breed, the identical phenotypes were not available, but the same region was associated with BCS and an alternative measure of insulin resistance. The closest gene was the poorly characterised *FAM174A*, which the authors sequenced. They found two closely linked variants, present in multiple breeds. The authors suggested an 11-guanine polymorphism near *FAM174A* might have potential as a predictive test for horses at risk of obesity/EMS/laminitis. Subsequently, an Australian group found no association with metabolic traits for that marker in 20 (non-Arabian) ponies [217]. Similarly, a larger study in multiple breeds, including Arabians, failed to replicate the BCS, laminitis, or insulin resistance associations, although assuming a dominant model did identify a significant association with the adipokine adiponectin [218].

Selective breeding can lead to enrichment of alleles within a breed, which are scarce in the wider population, meaning it is plausible that a genetic variant may exist in Arabians at this locus, genetically linked to the 11-guanine allele but which has yet to be identified. Thus, a real finding in Arabians might not be replicated in other breeds. However, replication in a larger, independent cohort of Arabians would be advisable. There is no evidence that this allele would be a suitable genetic test for obesity, EMS, or laminitis predisposition in other breeds.

Norton et al. [219] recently performed a GWAS focussed on EMS in a larger equine cohort (*n* = 550) representing two high-risk breeds (Welsh ponies and Morgan horses). By collecting rich metabolic data, they were able to test for association with multiple “endophenotypes”, more precise biochemical markers of insulin-related traits. Using endophenotypes may be more informative than performing a GWAS for a “convergent” phenotype such as laminitis, which can represent the clinical endpoint of multiple pathophysiological processes. The group appropriately adjusted for relevant confounding factors and for population stratification although not for BCS, meaning the results may identify loci associated with obesity rather than EMS per se. Hundreds of loci were identified across the multiple genome scans and the authors prioritised those which appeared to affect more than one breed.

The authors expressed surprise that there was not more overlap between breeds. That may be because major genetic determinants may be private to individual breeds, but it is perhaps more likely that the study was underpowered to find variants of small effect, or which were rare or invariate in one population. Protein coding genes in prioritised regions were enriched for involvement in pathways of inflammation, glucose metabolism, and lipid metabolism, all plausible as contributing to EMS pathology.

A final equine study considered the gene *HMGA2*, which had previously been identified as associated with short stature in pony breeds. In humans, short stature is a risk factor for insulin resistance. Concordantly, a study found pony breeds (which are smaller than horses) are more likely to get EMS [220]. The authors hypothesised a causal link between *HMGA2* and increased EMS susceptibility in ponies. They found a strong association with height in 264 Welsh ponies and a lesser association with several metabolic traits, which they reported as a pleiotropic effect.

## 10. From Humans to Animals and Back Again

### 10.1. Lessons for Animal Genetics

This review has summarised the wealth of research into the genetics of human obesity and its related metabolic perturbations, and the relative paucity of efforts to date in dogs, cats, and horses. Consequently, those studying veterinary species have much to learn from human geneticists and those studying laboratory animal models. By familiarising ourselves with those fields, there is much scope to fast-track animal studies to provide maximum insight into animal disease.

In particular, we note how careful delineation of clinical phenotypes has led to both genetic diagnoses and mechanistic insight in human patients, such as with familial partial lipodystrophy. Historically, such lean patients presenting with metabolic syndrome or T2D were considered inexplicable outliers. By recognising groups with shared clinical features, lipodystrophy has been recognised, patient care has improved, and we have a matured understanding of how healthy fat depots are essential to metabolic health [98]. Might similar clinical groupings exist, unrecognised, as underlying atypical presentations of EMS or determining why only some overweight cats become diabetic?

Human genetic epidemiological studies have, for some time, attempted to address the issue of cause and consequence between obesity and related pathologies. Performing GWAS not only for endpoints such as T2D or NAFLD, but also for those phenotypes corrected for BMI was a start in teasing out causal relationships between them, a process that has been improved with the advent of Mendelian randomisation [95]. Such approaches operate best at scale, which may limit their application in veterinary studies. Even so, an acute awareness that multiple routes can converge on a single phenotype (e.g., obesity, adipose dysfunction, and insulin signalling impairment converge to produce signs of metabolic syndrome) should inform our design and interpretation of veterinary GWAS.

Finally, human genetics also provides a lead in how to maximise the benefit of genetic studies. One notable contrast is that results of human genetic studies are rarely left “hanging”—mapped loci are further investigated, mouse models made to test the function of unknown genes, patients with specific genetic diagnoses are further studied, and precision treatments developed. This reflects a more established, larger, and better-funded research environment but provides a model for veterinary researchers to emulate.

Genetic findings have informed best practice to flag patients with uncommon presentations of common disease who may benefit from precision treatments [98,221]. In common, polygenic human disease, PRS are being used to counsel human patients about disease risk [80] and to stratify patients in research studies to understand better variable penetrance of variants of large effect [57]. In companion animals, there is a genetic test available for the retriever *POMC* mutation that can warn owners they have a dog at high risk of obesity and prompt them to effectively institute appropriate preventative measures [7], but other genetic findings in the field of obesity have yet to reach the veterinary clinic. Might we in future years detect cats with insulin signalling defects as candidates for insulin sensitising drugs in the pre-diabetic stage? Or might genetic profiling identify ponies at highest risk of laminitis? If so, genetic testing might prove a valuable clinical tool although one to be used only after robust validation of their utility in populations (e.g., breeds) different to that in which they were initially proposed.

### 10.2. Lessons from Animal Genetics

Fortunately, the flow of information between species need not be along a one-way street. Genetic studies of animal disease have already been informative to human research [7,222,223], and there is much potential to discover more. As genomic tools and better-annotated genomes become available, veterinary studies will be better able to provide insight into not just disease-associated loci but particular genes, mutations, and mechanisms too. That will clearly be a benefit to the species studied but also has the potential to benefit human health.

The current quality of genome builds and genomic tools available for companion animal studies vary by species. In dogs, the third-generation genome build (CanFam 3.1) has been available since 2014 [224], based on data from a Boxer dog, Tasha. Sequencing efforts integrating data from long read technologies produced genome sequences from a Great Dane (UMICH_Zoey_3.1, GCA_005444595.1, PacBio RSII) [225] and a basenji, (Basenji_breed-1.1, GCA_004886185.2, Sequel), meaning there are now three potential reference sequences available in the species. Canine geneticists were also the first to drive production of commercial companion animal genotyping arrays, with the earliest containing 49k SNP markers [226], later increasing to 173k, 220k (CanineHD BeadChip, Illumina, San Diego, CA, USA), and 710k densities (CanineHD Array, Thermofisher, Waltham, MA, USA). Today, there are available arrays that cover >1 million genetic markers (Canine Genotyping Array, Thermofisher, Waltham, MA, USA), a subset of which were selected for use on the 460k/670k arrays (Canine Genotyping Array A/B, Thermofisher, Waltham, MA, USA). Additionally, high quality imputation from a lower to higher density SNP array level and to genome sequencing density has been successful in dogs [227,228].

In cats, the newest genome build (Felis_catus_9.0) was made available in 2017 [184], and one relatively low-density genotyping array of 63k SNP markers exists [229] but is not commercially available. No use of imputation software in cats has been reported. In horses, the most recent genome build EquCab3.0 has been available since 2018 [230], and the first equine array was reported in 2014 at a density of 54k (Equine BeadChip, Illumina, San Diego, CA, USA) [231]. Nowadays, two more SNP arrays exist, both of which are commercially available, with marker densities of 65k (Equine BeadChip, Illumina, San Diego, CA, USA) and 670k (MNEc670K, Affymetrix, Santa Clara, CA, USA) [232]. Notably, a test equine array containing 2 million markers (MNEc2M) was created to inform the creation of MNEc670K and used successfully for imputation [232,233]. Imputation has also been performed in an equine population, from 54/65k density up to 670k [234].

Companion animal models of disease have the potential to “add value” to understanding broader biology for many reasons. Our animal companions share our homes and environments, spontaneously develop diseases similar to human conditions, for which they are often diagnosed and treated similarly but over a shorter time course. In some cases, disease processes that occur commonly in companion animals may be hard to study in humans. For instance, the retriever *POMC* mutation occurs in approximately 25% of Labrador retrievers, but the equivalent human mutations are very rare. The same molecule, β-MSH, is absent in rodents. Hence, in this situation, dogs provide an excellent animal model to shed light on a previously hard-to-illuminate area of human biology [7].

Similarly, cats are arguably more suitable than traditional animal models for the study of obesity associated T2D. Diabetes in cats bares closer resemblance to human T2D than that of the rodent model; it occurs naturally in cats, whereas T2D research in rodents is most often an induced disease state [235]. This means using the feline model of diabetes offers benefits for improved understanding of human T2D development, particularly for polygenic models [236,237].

Notably, there are advantages to studying such inbred populations, particularly dogs. Recent population bottlenecks at breed formation mean dogs have a very different genetic architecture to humans that makes complex trait mapping uniquely tractable in the species [238]. This means complex traits can be mapped in a breed with fewer individuals and fewer markers than in human populations. Dogs are therefore a compelling model for studying human metabolic disease [77,239,240,241]. Although, trait mapping in such inbred species means mapping to much larger loci than in humans [242,243], making causative variant identification within an associated locus more difficult.

Although cats do not display the same high level of LD as dogs, they have also been selectively bred and are proposed as potentially valuable models of human hereditary disease [183,244,245]. Horse genomes display lower levels of LD than both dogs and cats [183], but breed structure means there is potential for disease alleles being enriched and relatively easily studied within a discrete, definable population, meaning they too have potential to be valuable models of human disease [246].

## 11. Conclusions

The obesity epidemic is a major health concern in both human and companion animals, and there is a lot more to be discovered regarding the molecular basis of obesity and associated metabolic conditions. Despite clear evidence that obesity and related traits are highly heritable in companion animals, there are only limited studies to date investigating which genes are responsible and how they exert their effect. As this field matures, it promises tangible benefits for animal populations and, where considered as non-traditional animal models of obesity, has the potential to offer translational benefits too.

## Figures and Tables

**Figure 1 genes-11-01378-f001:**
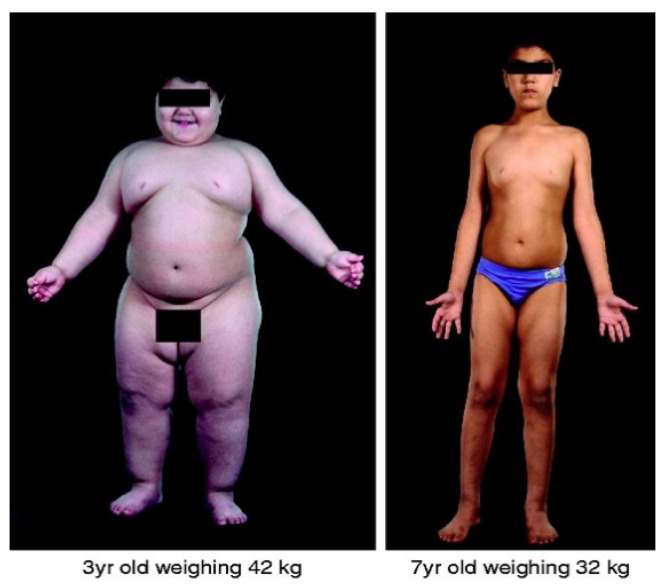
Effect of recombinant leptin treatment on child with congenital leptin deficiency. Photographs of a 3-year-old child before leptin treatment weighing 42 kg (**left**) and the same child weighing 32 kg (**right**) after 4 years of treatment with recombinant leptin therapy. Figure from Farooqi et al. [36].

**Figure 2 genes-11-01378-f002:**
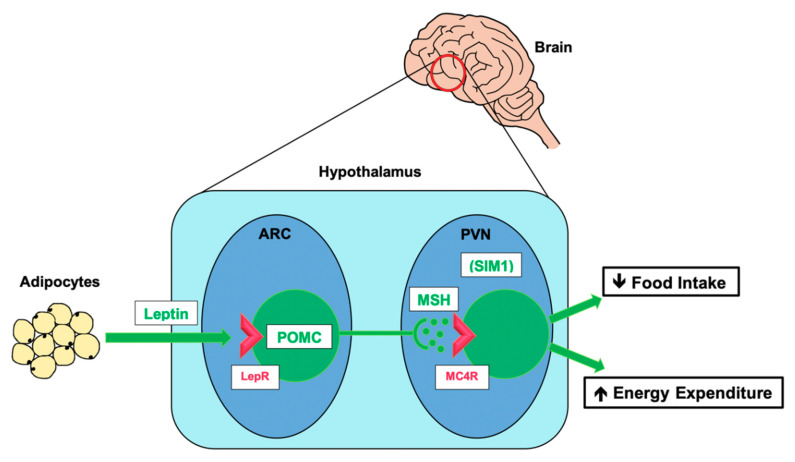
The leptin–melanocortin pathway. Simplified schematic of the leptin–melanocortin signalling pathway, which has a critical role in energy homeostasis by acting as a nexus through which information about energy status in the periphery can be relayed to the central nervous system (CNS) and integrated to control food intake and energy expenditure. ARC, arcuate nucleus of the hypothalamus; POMC, proopiomelanocortin; LepR, leptin receptor; PVN, paraventricular nucleus of the hypothalamus; MSH, melanocyte-stimulating hormone (α-MSH, β-MSH, γ-MSH); MC4R, melanocortin-4 receptor; SIM1, single-minded 1.

**Figure 3 genes-11-01378-f003:**
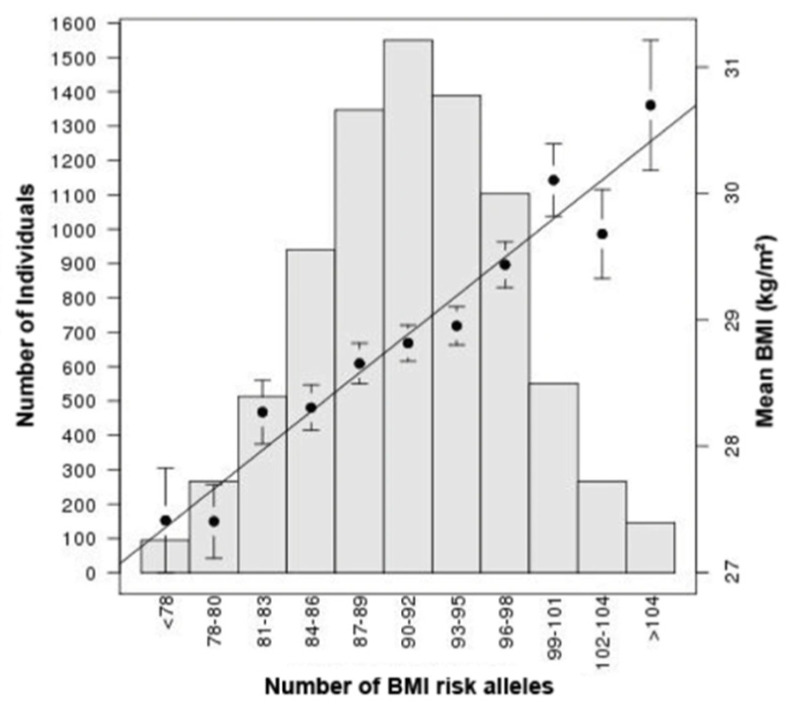
Histogram of cumulative effect of BMI risk alleles in a large human GWAS. Mean BMI for each bin is shown by the black dots (with standard deviation), corresponding to the right-hand y axis and is compared to simple PRS, based on unweighted risk allele counts. These data demonstrate how the cumulative effect of multiple polygenic risk alleles is strongly associated with BMI across the human population. Figure from Locke et al. [67].

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
