# Peer review of "The Genetic Basis of Obesity and Related Metabolic Diseases in Humans and Companion Animals"

_genes, 2020, doi:10.3390/genes11111378_

Round 1
Reviewer 1 Report
The authors present a comprehensive overview about the pathophysiology and genetics of obesity, first based on human and rodent studies, followed by the corresponding studies in companion species – dogs, cats and horses.
This is a well-written and thorough review about our current knowledge and status of research on the genetics of human and animal obesity. The authors have done excellent work with high value for future progression of the field.
I have just a few minor technical comments:
- Line 160: remove “HAVE”.
- Line 182: please explain the acronym WHR as “waist-to-hip ratio”.
- Line 183: “hundreds of OBESITY quantitative trait loci…”, remove “obesity”
- Line 203: should be “a large proportion OF genes”..
- Line 212: according to NCBI gene, FTO full name in mouse is “fat-mass and obesity-associated gene”, but for humans and dogs it is “FTO alpha-ketoglutarate dependent dioxygenase”. Please revise accordingly.
- Line 213: FTO should be in italics.
- Line 240: T2D needs explanation at the first appearance. Alternatively, this acronym can be presented in line 228 after “type 2 diabetes”.
- Line 341 : acronym “BCS” should appear after “body condition score”.
- Line 399: “Vulpes vulpes” should be in italics.
- Line 437: should be “investigated MC4R variants IN 270 dogs”.
- Line 444: Title for 7.1.3. should be “FTO”, thus consistent with other subtitles for specific genes. Alternatively, it should be “The FTO locus”.
- Line 445: It is not a good style to start a sentence with an acronym (FTO). Please use full name.
- Line 461: please refine the basis (cDNA, genomic, mRNA) for sequence homology between human and canine genes.
- Line 463: IRX3 should be in italics.
- Lines 482 and 483: same as for Lines 444 and 445. Also, please give the full name for INSIG2 at its first appearance.
- Lines 495 and 496: ibid
- Line 521: should be “that include leptin, …”.
- Line 570: please give full name for feline DM (this acronym has not been used with human Diabetes Mellitus descriptions above).
- Finally, I recommend the authors to provide information about all obesity candidate genes in different species in a concise tabular format including information about to which reference assembly the presented variants correspond.
Reviewer 2 Report
- To the authors :
- This paper proposes at first a detailed review of the definitions and genetic studies of comorbidities of obesity and metabolic syndrome in Humans, and then the authors review the similarities in obesity pathophysiology in 3 companion animals, dogs, cat and horses. Thus, we learn that the 3 animal models have a different role as a model of obesity and metabolic diseases in humans. It is not clear enough if the 3 animal species have sufficient genomic resources for complete, detailed and therefore interpretable genomic studies?
- T2D = type 2 diabete, I don’t think it is explained the 1st time.
- In the sentence « the ability to develop and maintain healthy fat stores appears to differ between individuals and across ethnic group » : Which ethnic groups show differences ?
- GWAS : The approach of whole genome association studies (GWAS) in dogs and cats is limited by the strong DL, so it is probably difficult to accurately identify loci containing causal mutations. Is this a serious limitation of these models? This could be discussed to make scientists aware that this is something to consider.
- When the authors talk about « The ‘missing heritability’ of obesity [61, 62] is hypothesised to be due to large effect rare variants yet to be identified, many loci of small/moderate effect too common to find with GWAS, non-additive genetic effects and copy number variants (CNVs), among others » I think the authors could discuss the current state of microbiome research in terms of understanding obesity, as diet plays a crucial role in shaping our gut ecosystem. I don’t ask for a review on microbiome and obesity, but a few words about the challenges of obesity-microbiome research would be helpful.
- Polygenic risk alleles : One of the most difficult aspects of using PRS is making sure that they are equally applicable to all and all ethnic groups in order to limit health disparities. Could it be that a PRS is limited to a single breed when studying dogs and cats?
